# CoCl_2_-Mimicked Endothelial Cell Hypoxia Induces Nucleotide Depletion and Functional Impairment That Is Reversed by Nucleotide Precursors

**DOI:** 10.3390/biomedicines10071540

**Published:** 2022-06-28

**Authors:** Barbara Kutryb-Zajac, Ada Kawecka, Alicja Braczko, Marika Franczak, Ewa M. Slominska, Roberto Giovannoni, Ryszard T. Smolenski

**Affiliations:** 1Department of Biochemistry, Medical University of Gdansk, Debinki 1 St., 80-211 Gdansk, Poland; ada.kaw.36@gmail.com (A.K.); alicja.braczko@gumed.edu.pl (A.B.); marika.franczak@gumed.edu.pl (M.F.); eslom@gumed.edu.pl (E.M.S.); 2Nutrafood Center and Department of Biology, University of Pisa, Via Derna 1, 56126 Pisa, Italy; roberto.giovannoni@unipi.it

**Keywords:** endothelium, hypoxia, nucleotides, adenine, ribose, nitric oxide, ATP, adenosine

## Abstract

Chronic hypoxia drives vascular dysfunction by various mechanisms, including changes in mitochondrial respiration. Although endothelial cells (ECs) rely predominantly on glycolysis, hypoxia is known to alter oxidative phosphorylation, promote oxidative stress and induce dysfunction in ECs. Our work aimed to analyze the effects of prolonged treatment with hypoxia-mimetic agent CoCl_2_ on intracellular nucleotide concentration, extracellular nucleotide breakdown, mitochondrial function, and nitric oxide (NO) production in microvascular ECs. Moreover, we investigated how nucleotide precursor supplementation and adenosine deaminase inhibition protected against CoCl_2_-mediated disturbances. Mouse (H5V) and human (HMEC-1) microvascular ECs were exposed to CoCl_2_-mimicked hypoxia for 24 h in the presence of nucleotide precursors: adenine and ribose, and adenosine deaminase inhibitor, 2′deoxycoformycin. CoCl_2_ treatment decreased NO production by ECs, depleted intracellular ATP concentration, and increased extracellular nucleotide and adenosine catabolism in both H5V and HMEC-1 cell lines. Diminished intracellular ATP level was the effect of disturbed mitochondrial phosphorylation, while nucleotide precursors effectively restored the ATP pool via the salvage pathway and improved endothelial function under CoCl_2_ treatment. Endothelial protective effects of adenine and ribose were further enhanced by adenosine deaminase inhibition, that increased adenosine concentration. This work points to a novel strategy for protection of hypoxic ECs by replenishing the adenine nucleotide pool and promoting adenosine signaling.

## 1. Introduction

Hypoxia plays a key role in the pathophysiology of cardiovascular diseases, including atherosclerosis, hypertension, and heart failure [1]. Due to their direct contact with circulating blood, vascular endothelial cells (ECs) are sensitive to any changes in oxygen levels [2]. Two main responses of ECs to insufficient oxygen availability have been described, which depend on the degree and duration of the insufficient oxygen supply. First, acute hypoxia stimulates ECs to release inflammatory mediators that promote neutrophil adhesion to ECs, or activate platelets [3]. Moreover, during early stages of hypoxia, adaptation and survival mechanisms are activated in response to decreased mitochondrial oxidative phosphorylation. These include shifting towards glycolytic metabolism, stimulating the production and activation of endothelial nitric oxide synthase (eNOS) or inducing angiogenic potential [4]. Second, when ECs are exposed to longer-term low oxygen tension, endothelial dysfunction and downregulation in eNOS expression become evident [5]. Sustained hypoxia also prompts ECs to generate factors that lead to vascular remodeling via the activation of smooth muscle cell proliferation and migration [6]. These are maladaptive responses that trigger significant vascular injury.

Intravascular ATP and adenosine are important regulators of vascular remodeling, endothelial barrier function, and neovascularization in numerous pathological states, including hypoxia, inflammation, and oxidative stress [7]. In response to pathological stimuli, vascular endothelium can release ATP, which triggers diverse signaling effects via ligand-gated (P2X) or G-protein-coupled (P2Y) receptors [8,9]. The following rapid cell-surface dephosphorylation of nucleotides turns off their signal transduction. This is of special importance, as adenosine is the product of nucleotide catabolism that binds to P1 receptors with opposite properties, frequently counteracting ATP and ADP effects [10]. The key ecto-enzymes that regulate the balance between extracellular nucleotides and adenosine comprise nucleoside triphosphate diphosphohydrolase-1 (known as CD39 or NTPDase1) and ecto-5′nucleotidase (E5NT or CD73) [11,12]. Extracellular adenosine signaling may be terminated by its uptake, via nucleoside transporters or further cell-surface catabolism by ecto-adenosine deaminase (eADA) [13]. 

Hypoxia-induced control of purine homeostasis and signaling has been revealed [7]. This included changes in cell-surface nucleotide catabolism and upregulated de novo purine synthesis, which may be the major components of an adaptive response of ECs to hypoxia and the loss of cellular ATP. Although it seems that, especially under chronic hypoxia conditions, these mechanisms do not provide an efficient restoration of endothelial ATP. Therefore, improving nucleotide status in ECs opens up new possibilities for pharmacological therapy of the vascular endothelium under disease conditions associated with prolonged oxygen deprivation. 

Cellular responses to low tissue oxygen levels are primarily controlled by transcription factors called hypoxia-inducible factors (HIFs) [14]. HIFs are heterodimer complexes, composed of inducible α and constitutively expressed β subunits [15]. Two α subunits (HIF-1α and HIF-2α) are known to accumulate in ECs, activating target genes that promote endothelial differentiation, growth, and migration [16]. It has been shown that HIFs are engaged in classic adaptive responses related to the induction of protective mechanisms that appears in acute hypoxia. However, as the duration of hypoxia continues, the activation of many other secondary signaling pathways can take place [17]. One of the best chemical models of HIF stabilization in vitro is hypoxia-mimetic agent, cobalt chloride (CoCl_2_), which prevents HIF degradation by hydroxylation [18]. As chronic hypoxia is a vital component in many cardiovascular diseases, this study aimed to investigate the effects of prolonged CoCl_2_ treatment on intracellular nucleotide concentration, extracellular nucleotide breakdown, mitochondrial function, and nitric oxide (NO) production in microvascular ECs, and to also investigate how nucleotide precursor supplementation and adenosine deaminase inhibition counteract the deleterious effects of CoCl_2_.

## 2. Materials and Methods

### 2.1. Cell Culture and Treatment

Mouse heart microvascular ECs (H5V) were kindly provided by Dr Patrycja Koszalka (Department of Cell Biology, faculty of Medical Biotechnology, Medical University of Gdańsk, Gdańsk, Poland). Human microvascular ECs (HMEC-1) were purchased from ATCC (USA). H5V cells were cultured in DMEM supplemented with 4.5 g/L glucose, 10% (*v*/*v*) fetal bovine serum (FBS), 1 mM sodium pyruvate, 2 mM glutamine and 1% (*v*/*v*) penicillin/streptomycin. HMEC-1 cells were cultured in MCDB131 medium with 10% (*v*/*v*) FBS, 10 ng/mL epidermal growth factor (EGF), 1 μg/mL hydrocortisone, 10 mM glutamine, and 1% (*v*/*v*) penicillin/streptomycin. Pig vascular endothelial cells (PIECs) were derived spontaneously from endothelial cell culture of a pig iliac artery and it was a gift from Dr. K. Welsh, Oxford, UK and transfected with human E5NT (hCD73-PIEC) and eNTPD1 genes (hCD39-PIEC), according to a method described earlier [19]. Mock transfected pig endothelial cells (MOCK-PIECs) were obtained by subjecting PIECs to the same [19] transfection procedure using empty vector. PIECs were cultured in RPMI supplemented with 10% (*v*/*v*) FBS, 2 mM L-glutamine, 1% (*v*/*v*) penicillin/streptomycin. All cells were cultured in a completely humid atmosphere at 37 °C and 5% CO_2_. The FBS was reduced to 1% 24 h before the experiment. Experiments were performed at 24-well plates on subconfluent monolayer ECs (80%) after the third passage. The cells were treated with a hypoxia-mimetic agent, 100 μM cobalt chloride (CoCl_2_) for 24 h at 37 °C and 5% CO_2_. Control cells were incubated with the vehicle (distilled water) under the same conditions. Adenine (SigmaAldrich, St. Louis, MO, USA) and ribose (SigmaAldrich, St. Louis, MO, USA) were added 45 min before the addition of CoCl_2_.

### 2.2. Quantification of Nucleotide and Their Catabolites Concentration Using RP-HPLC

After the treatment on 24 well plates, the cell medium was collected, centrifuged at 150× *g* for 3 min and the supernatant was used for the measurement of endogenous adenosine and inosine concentration using a previously described method [20]. The cell monolayer was then washed with PBS (pH 7.4) and ice-cold 0.4 mol/L HClO_4_ (300 μL) was added. After that, the plate was frozen at −80 °C and, 24 h later, it was thawed on ice and frozen again. After thawing, the supernatant was aspirated and 3 M K_3_PO_4_ was added to obtain the pH 6. All samples were centrifuged (20,800× *g*, 4 °C, 10 min) and the supernatants were used for the analysis of nucleotide concentrations with RP-HPLC [20]. The results were expressed as nmol/mg of protein.

### 2.3. Quantification of HIF1a Concentration in Cell Lysates

For the measurement of HIF1α concentration, cells were cultured and treated on 75 cm^2^ cell culture flasks. After the treatment, flasks were rinsed twice with PBS and cells were solubilized (1 × 10^7^ cells/mL) with a lysis buffer that contained 50 mM TRIS (pH 7.4), 300 mM NaCl, 25 mM NaF, 20 mM B-glycerophosphate, 3 mM EDTA, 1 mM MgCl_2_, 25 μg/mL leupeptin, 25 μg/mL pepstatin, 3.0 μg/mL aprotinin 10% (*w*/*v*) glycerol and 1% Triton X-100. Then samples were allowed to sit on ice for 15 min and frozen at −80 °C. Before use, samples were centrifuged (2000× *g* for 5 min) and, in supernatant, HIF1α concentration was measured using ELISA kit (cat no. DYC1935, RnDSystems, Minneapolis, MN, USA), according to the manufacturer’s instructions.

### 2.4. Quantification of Total Nitric Oxide, NO_2_^−^ and NO_3_^−^ Concentration in Cell Culture Medium

The concentration of total nitric oxide, nitrite, and nitrate in cell culture supernatants was measured using a Nitric Oxide (NO_2_^–^/NO_3_^–^) detection kit (cat no. ADI-917-010, Enzo LifeSciences, Farmingdale, NY, USA), according to the manufacturer’s instructions. 

### 2.5. Quantification of Cell Protein Content

After the experiments, cell protein from a 24-well plate was dissolved in 0.5 M NaOH and measured using the Bradford method, according to the manufacturer’s protocol. 

### 2.6. Measurement of the Extracellular ATP and AMP Hydrolysis, and Adenosine Deamination

After reaching confluence at 24-well culture plates, cells were washed with PBS and 1 mL of HBSS was added to each well. After that, 50 μmol/L of ATP, AMP, or adenosine were sequentially added and the samples were collected, after 0′, 5′, 15′, and 30′ of incubation at 37 °C, and analyzed using HPLC as described above. Protein residue was dissolved in 0.5 mol/L NaOH, and measured with the Bradford method. 

### 2.7. Immunofluorescence Staining

Immunofluorescent staining of endothelial total and phosphorylated at Ser1177 nitric oxide synthase (total-eNOS and phospho-eNOS) was performed in HMEC-1 cells plated in 96-well format (Corning, NY, USA). Cells were washed twice with FBS-free medium and incubated with rabbit primary anti-NOS antibody (Thermo Fisher Scientific, Waltham, MO, USA) for 1 h. After washing, Alexa Fluor 594- or 488-conjugated goat-anti-rabbit secondary antibody (Jackson Immuno, Cambridgeshire, UK) was added for 30 min. DAF-FM Diacetate (Thermo Fisher Scientific, Hertfordshire, UK) was used to detect nitric oxide. Alternatively, cells after the treatment were stained with MitoTracker Deep Red FM (Thermo Fisher Scientific, Hertfordshire, UK) at final concentration of 0.5 μmol/L for 45 min, then washed twice with PBS and fixed in methanol. Cell nuclei were counterstained by DAPI. Images were taken and analyzed using fluorescence microscope (Zeiss, Dresden, Germany) and ZEN software v.3.3 blue edition (Zeiss, Dresden, Germany).

### 2.8. Mitochondrial Function Analysis

The functional analysis of mitochondria was performed in HMEC-1 cells using a Seahorse XFp analyzer (Agilent, Santa Clara, CA, USA). Cells were seeded in XFp 8-well microplates (5 × 10^4^ cells/well) in a final volume of 80 μL MCDB full medium. The next day, cells were stimulated with/without CoCl_2_ for a further 24 h. Afterwards, cells were washed and incubated in 180 μL of Seahorse DMEM medium (enriched with 1 mM pyruvate, 2 mM glutamine, and 10 mM glucose) at 37 °C for 45 min. Oligomycin (inhibitor of ATP-synthase), carbonyl cyanide-p-trifluoromethoxyphenylhydrazone (FCCP; mitochondrial uncoupler), and a mix of rotenone (a complex I inhibitor) and antimycin A (a complex III inhibitor) were sequentially injected to a final concentration of 1.5 μM, 0.5 μM and 0.5 μM, respectively. Subsequently, the oxygen consumption rate (OCR) was recorded and basal respiration, ATP production, proton leak, maximal respiration, and nonmitochondrial respiration were calculated. The OCR was normalized to the protein concentration determined by the Bradford method.

### 2.9. Statistical Analysis

Statistical analysis was accomplished using InStat software (GraphPad, San Diego, CA, USA). Firstly, to assess the normality distribution, the normality tests were used. The comparisons of mean values between groups were assessed by unpaired Student’s t-test, or one-way Anova, followed by Holm-Sidak post hoc test, as appropriate. Pearson’s correlation coefficient was done to perform the correlation analysis. For each type of experiment, the exact value of n was provided. The significance was assumed at *p* ≤ 0.05. The error bars presented the standard error of the mean (SEM).

## 3. Results

### 3.1. Induction of Hypoxia-Mimic Conditions in Murine Cell Line H5V

First, we established a functional in vitro model of hypoxia-mimicked murine ECs. Treatment with CoCl_2_ for 24 h caused an increase in HIF-1α concentration in H5V cell lysate (Figure 1A). There were no significant differences in proliferation between hypoxic and control cells (Figure 1B,C). Hypoxia- mimicked H5V cells had decreased levels of intracellular ATP and ATP/ADP ratio, while intracellular NAD concentration remained unaffected (Figure 1D). To compare the activity of eNOS, we evaluated total concentrations of NO, NO_2_^–^ and NO_3_^–^ in cell culture media, which were all decreased after the treatment with CoCl_2_ (Figure 1E).

Next, we analyzed the effect of hypoxia on ecto-enzyme activities by quantifying the extracellular nucleotide conversion rates. A significant increase in extracellular ATP and AMP hydrolysis was detected in cells treated with CoCl_2_ for 24 h, in comparison to the vehicle-treated control (Figure 2A,B). Hypoxic cells also had a higher rate of extracellular adenosine deamination (Figure 2C).

### 3.2. Effects of Adenine and Ribose Supplementation on CoCl_2_-Treated Murine Cell Line H5V

The addition of 100 µM adenine and 2.5 µM ribose to the culture medium did not affect the proliferation of H5V cells under hypoxia-mimic conditions (Figure 3A,B) or in normoxia (Appendix AA). Although no significant changes were observed after the supplementation, regarding intracellular ATP/ADP ratio and NAD concentration, ATP concentration increased both in normoxic (Appendix AC) and hypoxic (Figure 3C) cells. The concentration of total NO and NO_2_^−^ was significantly higher in the culture media of cells treated with nucleotide precursors prior to induction of hypoxia-mimic conditions (Figure 3D).

### 3.3. Induction of Hypoxia-Mimic Conditions in Human Cell Line HMEC-1

CoCl_2_ treatment, via the HIF stabilization mechanism, also induced hypoxia-mimic conditions in human endothelial cell line HMEC-1. We verified this by measuring the concentration of HIF-1α, which was elevated 5 times after the stimulation (Figure 4A). CoCl_2_-mimicked hypoxia also decreased protein concentration in HMEC-1 (Figure 4B). The morphology of control and CoCl_2_-treated cells is shown in Figure 4C. Then, we analyzed the effect of CoCl_2_-treatment on mitochondria function in human ECs. It turned out that oxygen consumption rate was reduced in CoCl_2_-stimulated cells, and all mitochondria respiration parameters, including basal-, ATP-linked, and maximal respiration, as well as spare capacity, were decreased after CoCl_2_ (Figure 4D–F). This was related to inhibited transport of protons and, thereby, decreased mitochondrial membrane potential, analyzed by the staining with MitoTracker fluorescent dye that accumulates upon membrane potential (Figure 4G,H). Next, the effect of CoCl_2_ on the structure of mitochondria was investigated. Imaging of life cell mitochondria, using MitoTracker Deep Red, revealed mitochondrial aggregation after 24 h stimulation with CoCl_2_ (Figure 4H). CoCl_2_-challenged-HMEC-1 cells also displayed decreased concentration of intracellular ATP in comparison to control, while intracellular NAD concentration remained without significant changes (Figure 4I). Total NO concentration, as well as NO_2_^–^ and NO_3_^–^ levels, in cell culture media were decreased after CoCl_2_ treatment (Figure 4J). 

Next, we analyzed extracellular purine metabolism by quantifying the rates of exogenous nucleotide and adenosine conversions. We observed higher rates of ATP and AMP hydrolysis in CoCl_2_-treated cells (Figure 5A). There was no significant change in adenosine deamination rate between cells under normoxia and hypoxia-mimic conditions. However, a trend toward higher ecto-adenosine deaminase activity was observed (Figure 5A). This extracellular nucleotide metabolism pattern was reflected by slightly higher endogenous adenosine concentration in the cell media of CoCl_2_-treated cells, but it was rapidly deaminated to inosine (Figure 5B). The use of adenosine deaminase inhibitor, deoxycoformycin (dCF), together with CoCl_2_, counteracted extracellular adenosine degradation, improving its level in cell medium that was at least 5 times higher than in CoCl_2_-treated cells alone (Figure 5B). To determine the relation between adenosine and NO levels in EC media, we investigated the capacity of pig endothelial cells, transfected with human ecto-nucleotidases (hCD39 and hCD73), to NO production. Firstly, we characterized transfected cells that exhibited a massive, 60 times higher rate of ATP hydrolysis (hCD39-PIEC) and AMP hydrolysis (hCD73-PIEC), in comparison to mock-transfected control cells MOCK-PIEC (Figure 5C). This resulted in moderately higher endogenous adenosine levels in hCD39-PIEC cell media, while hCD73-PIEC revealed highly increased levels of both endogenous adenosine and inosine concentrations in cell culture media (Figure 5D). The adenosine concentration positively correlated with NO concentration in the PIEC cell media (Figure 5E). However, there were no differences in the ability to produce NO between MOCK- and hCD39-PIECs, while hCD73-transfected cells produced much more NO than control and hCD39 cells (Figure 5E).

### 3.4. Effects of Adenine and Ribose Supplementation on CoCl_2_-Treated Human Cell Line HMEC-1

Similar to H5Vs, HMEC-1 cells were supplemented with nucleotide precursors, adenine and ribose, before CoCl_2_ treatment. No significant changes between supplemented and non-supplemented hypoxia-mimic cells were observed regarding cell proliferation (Figure 6A,B). Supplementation increased intracellular ATP concentration, although it did not significantly affect intracellular ATP/ADP ratio and NAD concentration (Figure 6C). There were tendencies toward higher concentrations of total NO and NO_3_^−^ in adenine and ribose supplemented HMEC-1 cells under hypoxia-mimic conditions (Figure 6D). Moreover, quantitative analysis of mean fluorescence intensity (MFI) for total-eNOS, phospho-eNOS (Ser1177) and NO staining showed that CoCl_2_ treatment significantly decreased total- and phospho-eNOS expression, as well as phospho-eNOS/total-eNOS ratio, providing the evidence of subsequent loss of eNOS activity (Figure 6E–H). Adenine and ribose supplementation in part reversed this effect, resulting in no significant difference between this group and control cells in the context of total-eNOS expression and NO MFI (Figure 6E,F). 

### 3.5. Effects of Nucleotide Precursors and Adenosine Deaminase Inhibition on CoCl_2_-Treated Human Cell Line HMEC-1

Then we analyzed the joint effect of nucleotide precursor supplementation and adenosine deaminase inhibition on functional impairment of ECs stimulated with CoCl_2_. Combined treatment of adenine and ribose with dCF, before CoCl_2_-mimicked hypoxia, revealed higher total NO concentration in the culture media than in non-treated cells (Figure 7). The use of adenosine receptor antagonists demonstrated that at least part of this effect is dependent on extracellular adenosine signaling. The addition of DPSPX, an A1 adenosine receptor antagonist, reversed the influence of nucleotide precursor and adenosine deaminase inhibitor treatment. While the usage of A2a and A2b adenosine receptor antagonists did not affect the protective effects of our supplementation (Figure 7).

## 4. Discussion

This work demonstrated for the first time that combined application of the precursors for the nucleobase reutilization pathway, together with adenosine deaminase inhibitor, reversed nucleotide depletion and endothelial impairment induced by CoCl_2_-mimicked chronic hypoxia. The supplementation with adenine and ribose augmented ATP concentration and improved NO production in CoCl_2_-treated mouse and human ECs. Moreover, the termination of adenosine deaminase-mediated metabolism straightened the protective effects of nucleotide precursors on ECs, enhancing the bioavailability of adenosine to its signaling via the A1 receptor.

The most common model to induce hypoxia in vitro is the reduction of O_2_ concentration to the level of 1–5% in the culture environment [18]. However, the major limitation of hypoxic chambers is that the local O_2_ concentration does not necessarily correspond at the cellular level [21]. Thus, we decided to mimic hypoxia using CoCl_2_, that, by competing with iron ions, inhibits the prolyl hydroxylase domain responsible for HIF-1α hydroxylation, and, therefore, enables its stabilization [18]. In this study, 24 h treatment with CoCl_2_ increased HIF-1α concentration in cell lysates in mouse H5Vs and human HMEC-1 ECs. In analyzed endothelial cells, hypoxia-mimic conditions led to intracellular ATP depletion and decreased ATP/ADP ratio that was the effect of markedly diminished mitochondrial oxidative phosphorylation. This was associated with inhibited transport of protons and, thereby, decreased mitochondrial membrane potential in CoCl_2_-treated cells. 

It was believed that ECs are mainly glycolysis-dependent for energy supply [22], but increasing data demonstrates that mitochondrial function is essential in the homeostatic regulation of endothelium [23,24]. There is evidence that proliferative ECs increasingly rely on mitochondrial oxidative phosphorylation and, particularly, on the mitochondrial proton gradient [25]. Moreover, it has been shown that key metabolic glucose pathways clearly modulate endothelial inflammation in in vitro and in vivo studies [26]. The enhancement of mitochondrial respiration or the pentose phosphate pathway had counterregulatory anti-inflammatory effects, whereas NF-kB-PFKFB3 signaling-mediated glycolysis stimulation triggered inflammation in [26]. Also, as in our study, when hypoxia occurs, the mitochondrial respiration in ECs decreases. Hence, especially during an acute hypoxia condition, glycolytic metabolism is augmented to compensate for the reduction in ATP regeneration [27]. ECs are prompted towards glycolytic metabolism under hypoxic conditions via the activation of HIF1α, which induces the expression of pyruvate dehydrogenase kinase 1 (PDK1) and prevents the conversion of pyruvate to acetyl-CoA [28]. Therefore, as acetyl-CoA is the principal substrate for tricarboxylic acid (TCA) cycle, HIF1α emphasizes the reliance of ECs on glycolysis by inhibiting TCA-induced oxidative phosphorylation [16]. However, as intracellular ATP concentration was still diminished in our experimental setting of prolonged hypoxia, it seems that glycolysis metabolism was not fully sufficient to regenerate the ATP pool. This could trigger a series of changes in endothelial cell function [29], including decreased production of NO by eNOS as we observed in both H5Vs and HMEC-1 cells after CoCl_2_ treatment [30]. 

NO is an important protective molecule in the vasculature as it acts as a critical vasodilator and inhibitor of many pathological processes, including platelet aggregation and adhesion, leukocyte adhesion to vessel walls, mitogenesis, or proliferation of smooth muscle cells [31]. Most of the vascular NO is produced by eNOS, so an imbalance in eNOS activity is characterized as endothelial dysfunction [32]. ENOS is a Ca^2+^-dependent enzyme activated by intracellular Ca^2+^ signaling [33]. In non-stimulated ECs, spontaneous Ca^2+^ release, via inositol phosphate receptors (IP_3_), maintains a basal level of NO, which opposes the vascular tone [34]. Chemical and mechanical stimuli increase this basal Ca^2+^ entry, to stimulate NO production and promote vasodilation [35]. It has been shown that mitochondrial regulation of Ca^2+^ signals was controlled by ATP production that, when reduced by ATP synthase inhibition or mitochondrial depolarization, abolished local IP_3_-mediated Ca^2+^ release [36]. Two mechanisms can elucidate how mitochondrial-derived ATP controls Ca^2+^ signaling by IP_3_ receptors. First, high ATP concentration is required to generate IP_3_ and its precursors via phospholipase C and phosphoinositide kinase [37,38], and, second, ATP may sensitize the channel to promote Ca^2+^ release by binding specifically to an ATP-binding site on the IP_3_ receptor [39]. Therefore, ATP production, via mitochondrial respiration, is a key component that provides long-term regulation of endothelial signaling by IP_3-_triggered local Ca^2+^ release in native endothelium. 

An increasing line of evidence suggests that acute and chronic hypoxia also affects eNOS function in a Ca^2+^-independent manner. Short-term oxygen deprivation via HIF-1α signaling upregulates adaptive NO production by ECs [40], whereas longer-term hypoxic stimulation leads to decreased eNOS activity [5]. A significant role in the regulation of eNOS activity is played by the phosphorylation state of specific serine, threonine and tyrosine residues. It has been shown that acute hypoxia that lasts below 3 h stimulates eNOS phosphorylation at Ser-1177 via the PI3K-Akt pathway and, thus, increases eNOS activity and NO release [41]. In this study, we showed that CoCl_2_-stimulated ECs revealed decreased eNOS phosphorylation (Ser1177) and phospho-eNOS/total-eNOS ratio, which rather suggested that 24 h CoCl_2_ treatment displayed similar effects on eNOS activity as chronic hypoxia. Although, adenosine and ribose supply did not change phospho-eNOS/total-eNOS ratio under our hypoxia-mimic conditions.

In human ECs, CoCl_2_-mimicked hypoxia also affected protein concentration, which could be due to the effect on cell proliferation as a result of increased necrosis or apoptosis. Therefore, we analyzed how CoCl_2_ affects mitochondrial structure as disturbances in mitochondrial morphology are linked with the cell death [42]. Imaging of endothelial cell mitochondria using MitoTracker revealed their aggregation after CoCl_2_ treatment. It has previously been shown that mitochondria become aggregated after exposure to apoptotic stimuli, followed by cytochrome c release that leads to caspase-3-dependent apoptosis [43].

Besides affecting endothelial cell metabolism and intracellular nucleotide levels, the hypoxia-driven response significantly alters purinergic signaling pathways. Oxygen deprivation stimulates ECs to release ATP that triggers vascular inflammation, endothelial permeability, and thrombosis via P2 receptor stimulation [9,44]. These pathological effects of soluble nucleotides can be suppressed by redirecting them to extracellular hydrolysis via the CD39-CD73 axis [45]. In the cardiovascular system, the endothelial ecto-nucleotidases play a key role in the balance between nucleotides and protective adenosine, maintaining and strictly regulating the homeostatic process and counteracting excessive vascular leakage, or clot formation [46,47]. In our work, CoCl_2_ treatment moderately activated ATP breakdown into adenosine on the surface of H5V cells but had a strong effect on extracellular ATP and AMP hydrolysis in human HMEC-1 cells. This was in line with other studies, where up-regulation of endothelial NTPDase1/CD39 and ecto-5′nucleotidase/CD73 under oxygen deprivation has been shown [48,49]. Thus, in hypoxic conditions the levels of extracellular adenosine could rise substantially, as an early protective response against excessive cellular damage. However, adenosine concentration in the extracellular environment is also controlled by its degradation through cell-surface enzymes and by transmembrane transport processes [50,51]. In our work, we observed augmented adenosine to inosine catabolism on the surface of CoCl_2_-treated endothelial cells. Similarly, other reports have shown induction of eADA activity in endothelial cells cultured in a hypoxia chamber [13]. This could be an adaptive metabolic mechanism to chronically elevated adenosine levels during hypoxia, which turns off adenosine-dependent protective pathways. 

Pharmacological replenishment of intracellular ATP is a promising strategy to improve energy-consuming cell functions, including the synthesis and secretion of endothelium-derived factors or transport functions [52]. Rebuilding the ATP pool in ECs is also important, due to its role in the production of extracellular adenosine that, via receptor signaling, further optimizes endothelial function [53]. One of the options to increase the concentration of ATP in the ECs is to stimulate mitochondrial respiration as a more efficient way to produce energy than glycolysis. It has been shown that resveratrol increased mitochondrial content in ECs. This was related with SIRT1 activation, further eNOS upregulation, and induction of mitochondrial biogenesis [54]. Resveratrol treatment also stimulated biogenesis of mitochondria in type 2 diabetic mice aortas [54]. However, manipulating the mitochondrial function may also be unfavorable in ECs, due to mitochondrial ROS (mROS) production [55]. At relatively low levels, mROS are critical to support normal or compensatory functions of the cell [56]. For instance, cells use a severe increase in mROS for HIF1α stabilization during an hypoxia condition and afterwards restrain ROS production in chronic hypoxia to avoid cellular damage [57]. However, in some pathological conditions, such as hyperglycemia, excessive production of mROS occurs and it is implicated in glucose-mediated vascular damage [58]. The key mechanism by which mROS is involved in endothelial dysfunction is eNOS uncoupling [59]. This phenomenon is based on uncoupling eNOS from its cofactor or substrate leading to the production of superoxide (O_2_^•−^) instead of NO. Moreover, O_2_^•−^ further reacts with NO to form ONOO^−^, thus triggering oxidative stress and further endothelial dysfunction and vascular disease [60].

Another strategy for replenishing the intracellular ATP pool in ECs is to reinforce its re-synthesis [61]. The restoration of purine nucleotides can be accomplished from precursors, such as amino acids, formate or bicarbonate, by the multipart de novo pathway or by salvage pathways from purine derivative precursors, bases and nucleosides [62]. In physiology, de novo purine synthesis plays a minor role in ECs in comparison to salvage pathways [62]. However, it has been shown that a contribution of de novo synthesis becomes more efficient under pathophysiological conditions, such as occurs during acute hypoxia [7]. It should be emphasized that the stimulated pathways of de novo nucleotide synthesis take place at considerable energy cost, exhausting cellular energy reserves [63]. Therefore, to re-synthetize nucleotides under chronic hypoxia conditions, we supplemented ECs with adenine and ribose, key substrates for adenine nucleotide reutilization pathway [63]. Consequently, both analyzed types of microvascular ECs represented higher intracellular ATP concentrations after the treatment, which correlated with the improved endothelial function. We demonstrated previously that adenine supply protected ECs and cardiomyocytes from ATP loss caused by adenosine kinase (ADK) inhibition [64]. It has been demonstrated that during hypoxia, ADK activity also decreases in ECs, shunting endothelial adenosine from the salvage pathway to intravascular release [65]. This underlines the importance of using adenine and ribose for the replenishment of ATP in ECs. However, as we have shown using human and porcine endothelial cells, adenosine released into the extracellular space has a very short half-life due to its catabolism by eADA, which limits its protective receptor effects [66]. Therefore, we proposed to inhibit adenosine to inosine catabolism by 2′deoxycofromycin, a cell-penetrated inhibitor of both intracellular ADA and cell-surface eADA [67]. Consequently, we observed that combined treatment with adenine, ribose, and 2′deoxycoformycin led to increase in NO production under chronic CoCl_2_ stimulation and this effect was reversed upon administration of the adenosine A1 receptor antagonist. We have previously shown that three adenosine receptor subtypes, A1, A2a and A2b, are abundantly expressed in human endothelial cells [51]. Using specific agonists and antagonists, we revealed that A1 and A2b receptor stimulation protected the endothelium from triggering the inflammatory phenotype, to a similar extent as eADA inhibition by dCF [51]. Such conclusions can also be drawn from this work, as using porcine cells transfected with human E5NT, we revealed that the increased extracellular adenosine concentration positively correlated with NO production by these cells. Although after adenine, ribose and dCF treatment of HMEC-1 cells, we did not observe differences in eNOS phosphorylation. Therefore, we assume that NO release stimulated via A1R signaling is rather dependent on PLC/IP3/Ca^2+^ pathway, as we described on Figure 8, but this aspect requires further investigation.

Summarizing, this study emphasized the importance of modulating intra- and extra-cellular adenosine metabolism (Figure 8), which, together with replenishing its production from nucleotides, is crucial for endothelial function and cardiovascular homeostasis. 

## 5. Conclusions

Nucleotide precursor supplementation prevented pathological effects of hypoxia on endothelial cells. This intervention corrected disrupted intracellular nucleotide status as well as endothelial cell function. The protective effects of adenine and ribose on endothelial cells were further targeted by adenosine deaminase inhibition, enhancing adenosine receptor signaling. This highlights the benefit of pharmacological modifications that increase the nucleotide pool in endothelial cells under disease conditions associated with chronic oxygen deprivation. 

## 6. Highlights

CoCl_2_-mimicked hypoxia disrupted nucleotide pool and function in mouse and human microvascular endothelial cells.The precursors for the reutilization pathway, adenine and ribose rescued the intracellular nucleotide pool and increased nitric oxide production in CoCl_2_-treated endothelium.The use of adenosine metabolism inhibitor, 2′deoxycoformycin, together with nucleotide precursors, further improved endothelial function in CoCl_2_-mimicked hypoxia.

## Figures and Tables

**Figure 1 biomedicines-10-01540-f001:**
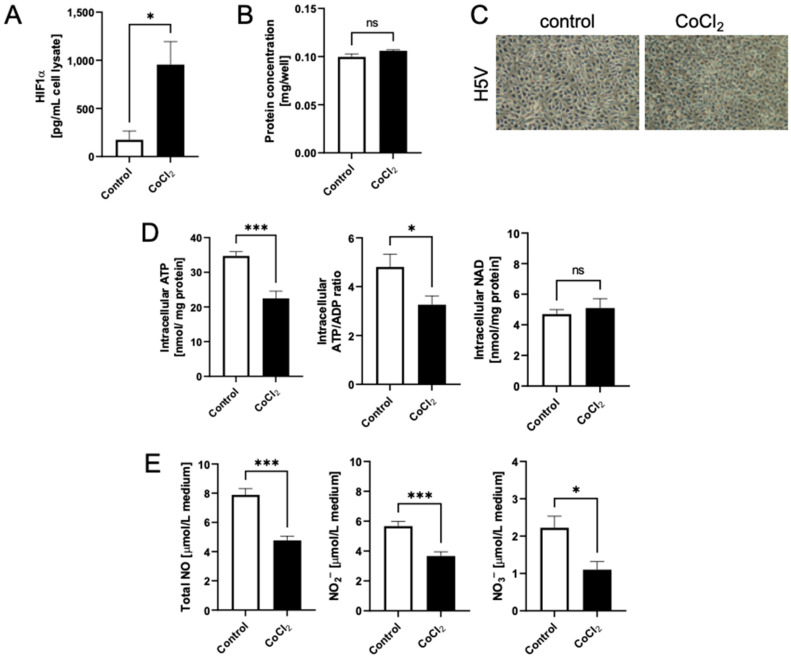
CoCl_2_-mimicked hypoxia depleted intracellular ATP concentration and endothelial cell function in murine heart endothelial cells (H5Vs). (**A**). HIF-1α levels were determined in H5V cell lysate in control cells and following 24 h stimulation with 100 μM CoCl_2_; (**B**). Total cell protein, (**C**). Representative H5V cell images, (**D**). Intracellular ATP and NAD concentration, (**E**). The concentration of total NO, NO_2_^–^ and NO_3_^–^ in the cell culture media of control cells and after 24 h of hypoxia-mimic conditions. Results are shown as mean ± SEM; *n* = 6; * *p* < 0.05; *** *p* < 0.001; ns = not significant.

**Figure 2 biomedicines-10-01540-f002:**
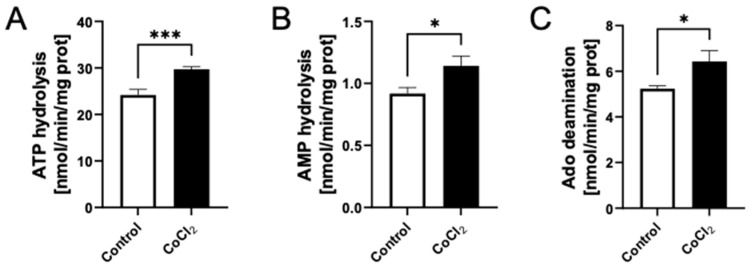
CoCl_2_-mimicked hypoxia promoted extracellular nucleotide hydrolysis and adenosine deamination in H5V endothelial cells. The rates of extracellular ATP hydrolysis (**A**), AMP hydrolysis (**B**), and adenosine (Ado) deamination (**C**) in H5V cells in control and hypoxia-mimic conditions (CoCl_2_). Results are shown as mean ± SEM; *n* = 6; * *p* < 0.05, *** *p* < 0.01.

**Figure 3 biomedicines-10-01540-f003:**
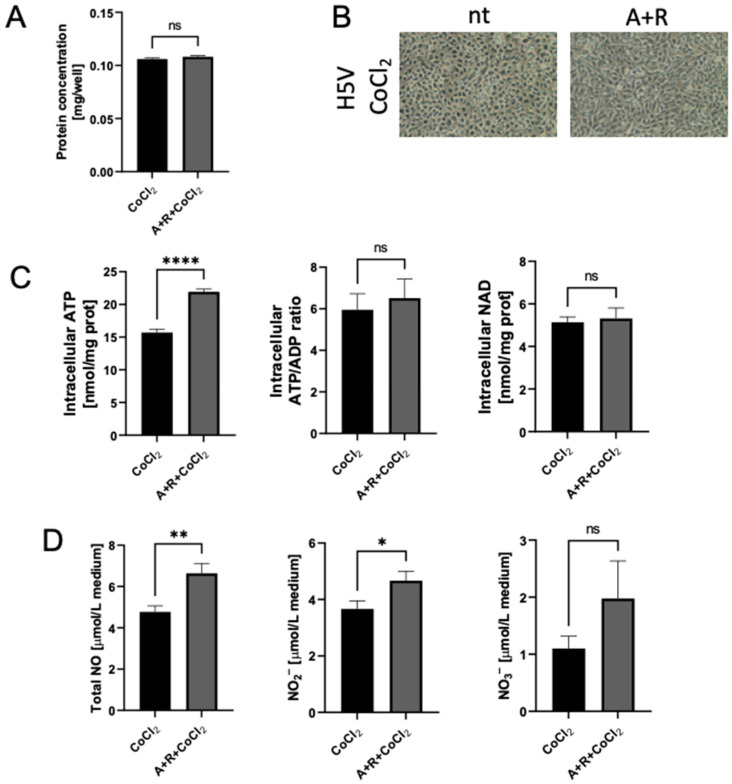
Adenine and ribose supplementation enhanced intracellular nucleotide pool and endothelial function in CoCl_2_-treated H5V endothelial cells. (**A**). Total cell protein, (**B**). Representative H5V cell images after CoCl_2_-mimicked hypoxia in the presence (A + R) and absence (nt) of 100 μM adenine and 2.5 μM ribose; (**C**). Intracellular nucleotide concentration after A + R treatment during hypoxia conditions. (**D**) The concentration of total NO, NO_2_^–^ and NO_3_^–^ in the cell culture media after adenine and ribose treatment (A + R) after 24 h of CoCl_2_ treatment. Results are shown as mean ± SEM; *n* = 6; * *p* < 0.05; ** *p* < 0.01; **** *p* < 0.0001; ns = not significant.

**Figure 4 biomedicines-10-01540-f004:**
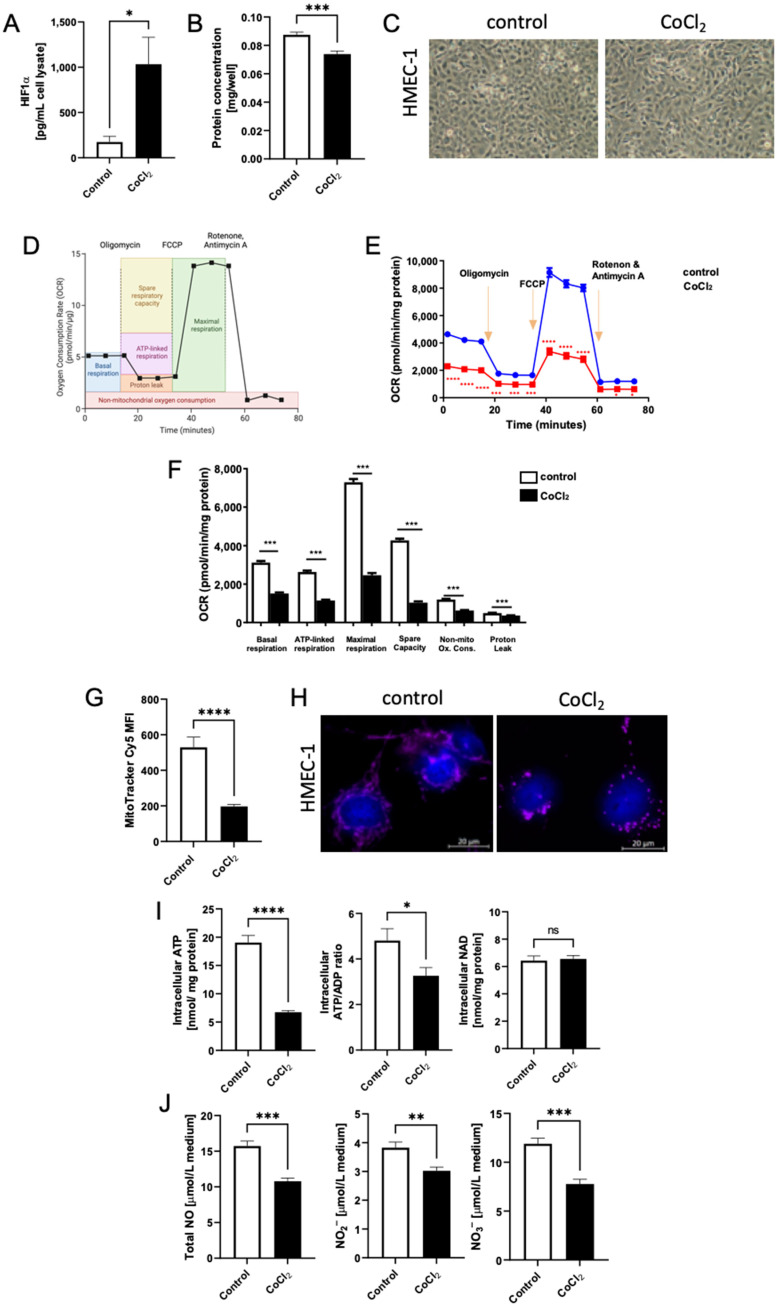
CoCl_2_-mimicked hypoxia depleted intracellular ATP concentration via decreased oxidative phosphorylation that was related to disrupted endothelial function in human microvascular endothelial cells (HMEC-1s). (**A**). HIF-1 α levels were determined in HMEC-1 cell lysate in control cells and following 24 h stimulation with 100 μM CoCl_2_; (**B**). Total cell protein, (**C**). Representative HMEC-1 cell images; (**D**). Scheme of MitoStress Test created with https://BioRender.com (accessed on 27 April 2022); (**E**). Oxygen consumption rate (OCR) as studied using the Seahorse XFp8 Extracellular Flux Analyser after the sequential addition of oligomycin, 2-[2-(3-Chlorophenyl) hydrazinylyidene] propanedinitrile (FCCP), rotenone and antimycin; (**F**). Calculated mitochondrial function parameters in control cells and after 24 h of CoCl_2_ treatment; (**G**). Quantitative analysis of mean fluorescence intensity (MFI) for MitoTracker Deep Red and (**H**); Representative images presenting fluorescence staining with MitoTracker Deep Red (magenta), and nuclei (blue) in control cells and after 24 h of CoCl_2_ treatment; (**I**). Intracellular ATP and NAD concentration; (**J**). The concentration of total NO, NO_2_^–^ and NO_3_^–^ in the cell culture media of control cells and after 24 h of CoCl_2_ treatment. Results are shown as mean ± SEM; *n* = 6; * *p* < 0.05, ** *p* < 0.01, *** *p* < 0.001, **** *p* < 0.0001; ns = not significant.

**Figure 5 biomedicines-10-01540-f005:**
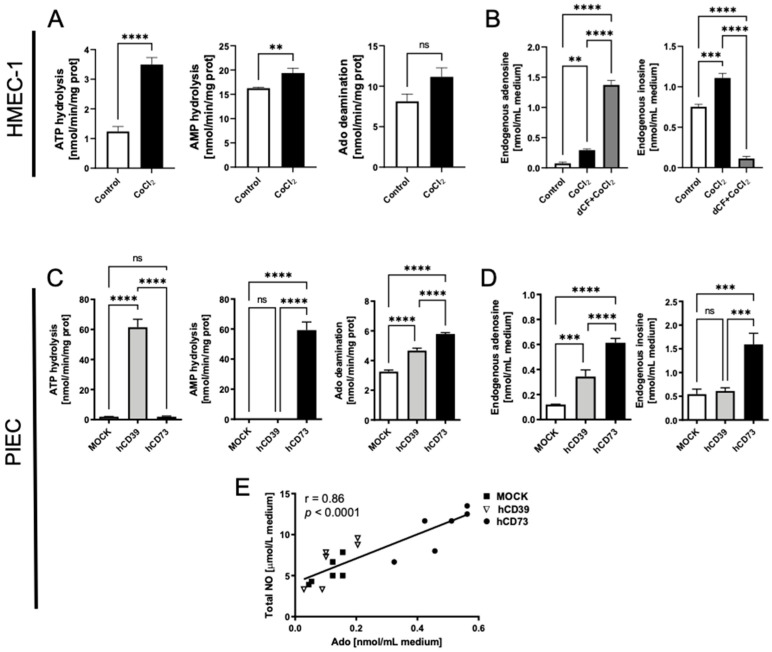
CoCl_2_-mimicked hypoxia upregulated extracellular nucleotide hydrolysis and adenosine deamination in human endothelial cells, while the inhibition of adenosine catabolism maintained its extracellular levels at those corresponded with enhanced NO release, as verified using porcine cells transfected with human ecto-nucleotidases. (**A**). The rates of extracellular ATP hydrolysis, AMP hydrolysis, and adenosine (Ado) deamination in control and CoCl_2_-treated cells. (**B**). Endogenous Ado and inosine (Ino) concentration in the medium of control and CoCl_2_-treated cells in the presence of absence of adenosine deaminase inhibitor, 2′deoxycoformycin (dCF, 10 μM). (**C**). The rates of extracellular ATP hydrolysis, AMP hydrolysis, and Ado deamination in mock-transfected porcine endothelial cells (MOCK-PIEC) and PIEC transfected with human CD39 (hCD39-PIEC) and CD73 (hCD73-PIEC). (**D**). Endogenous Ado and Ino concentration in the medium of MOCK-PIEC, hCD39-PIEC and hCD73-PIEC. (**E**). Pearson’s correlation of total nitric oxide (NO) vs. Ado concentration in the medium of MOCK-PIEC, hCD39-PIEC and hCD73-PIEC. Results are shown as mean ± SEM; *n* = 6; ** *p* < 0.01, *** *p* < 0.001, **** *p* < 0.0001; ns = not significant.

**Figure 6 biomedicines-10-01540-f006:**
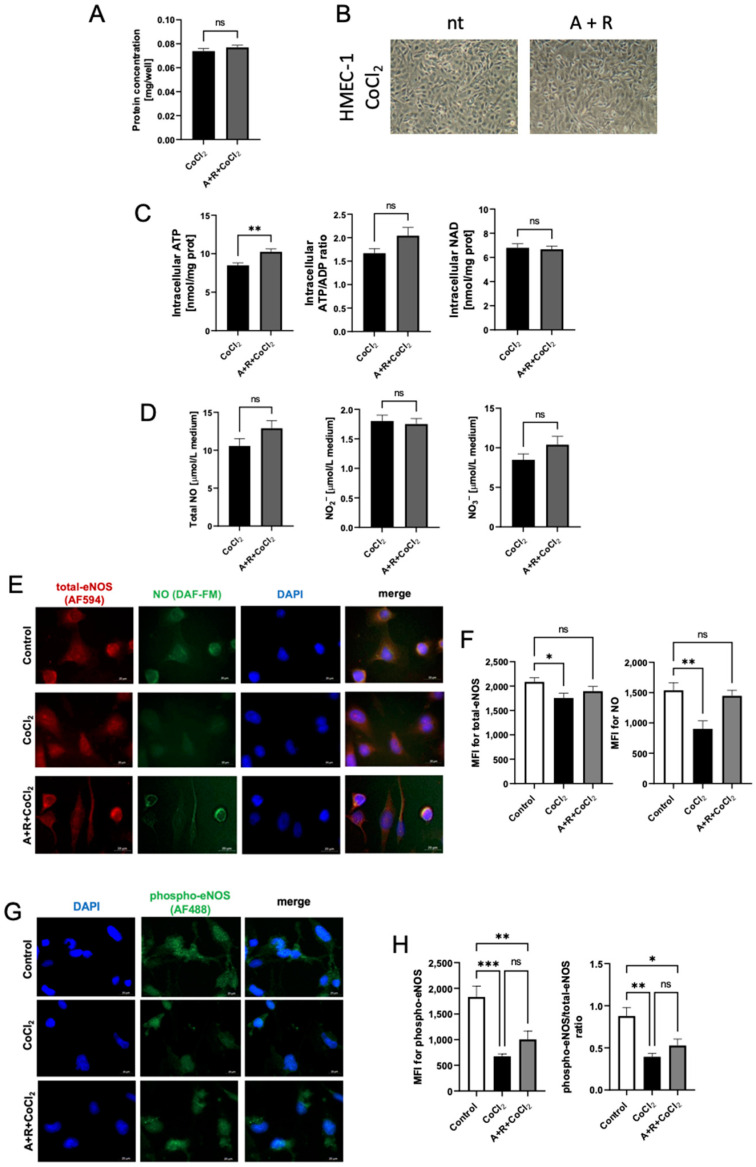
Adenine and ribose supplementation improved intracellular nucleotide status and function of HMEC-1 cells under hypoxia-mimic conditions. (**A**). Total cell protein, (**B**). Representative HMEC-1 cell images after CoCl_2_ stimulation in the presence (A + R) and absence (nt) of 100 μM adenine and 2.5 μM ribose; (**C**). Intracellular nucleotide concentration after A + R treatment during hypoxia-mimic conditions. (**D**) The concentration of total NO, NO_2_^–^ and NO_3_^–^ in the cell culture media after adenine and ribose treatment (A + R) after 24 h of CoCl_2_ treatment. (**E**). Representative images presenting fluorescence staining of total endothelial nitric oxide synthase (total-eNOS, red), nitric oxide (NO, green), and nuclei (blue). (**F**). Quantitative analysis of mean fluorescence intensity (MFI) for total eNOS and NO staining in control cells and after CoCl_2_ treatment in the presence and absence of A + R supplementation. (**G**). Representative images presenting fluorescence staining of phosphorylated endothelial nitric oxide synthase at Ser 1177 (phospho-eNOS, green), and nuclei (blue). (**H**). Quantitative analysis of MFI for phospho-eNOS and phospho-eNOS/total-eNOS ratio in control cells and after CoCl2 treatment in the presence and absence of A + R supplementation. Results are shown as mean ± SEM; *n* = 6; * *p* < 0.05, ** *p* < 0.01, *** *p* < 0.001; ns = not significant.

**Figure 7 biomedicines-10-01540-f007:**
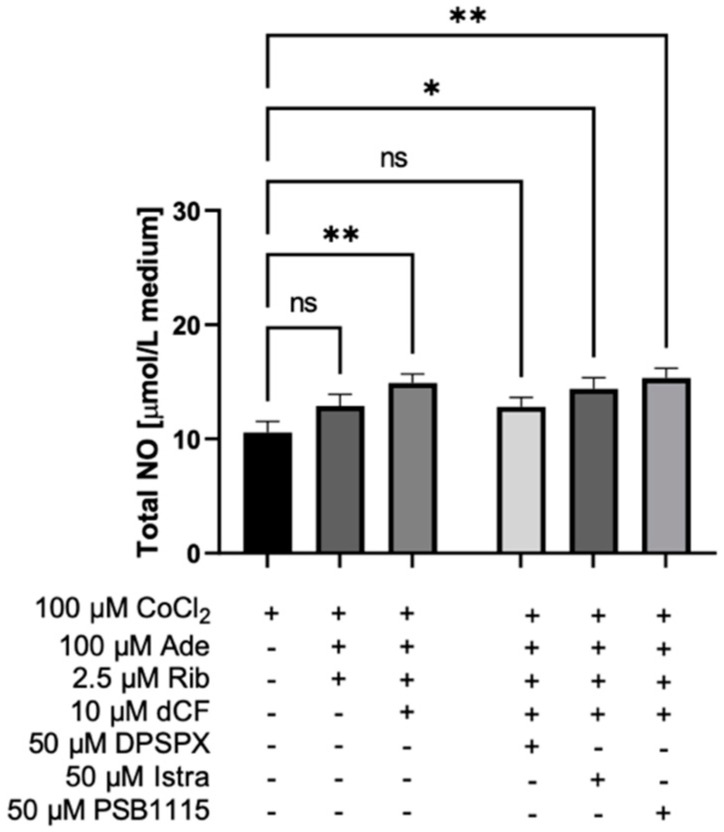
Adenosine deaminase inhibition together with adenine and ribose supplementation provided benefit for endothelial cell function in CoCl_2_-mimicked hypoxia by A1 adenosine receptor mediated mechanism. The concentration of total NO in the HMEC-1 cell culture media after 24 h of CoCl_2_-mimicked hypoxia in the presence of adenine (Ade), ribose (Rib), 2′deoxycoformycin (dCF, adenosine deaminase inhibitor), DPSPX (A1R antagonist), istradefylline (Istra, A2aR antagonist) and PSB1115 (A2bR antagonist). Results are shown as mean ± SEM; *n* = 6; * *p* < 0.05, ** *p* < 0.01; ns = not significant.

**Figure 8 biomedicines-10-01540-f008:**
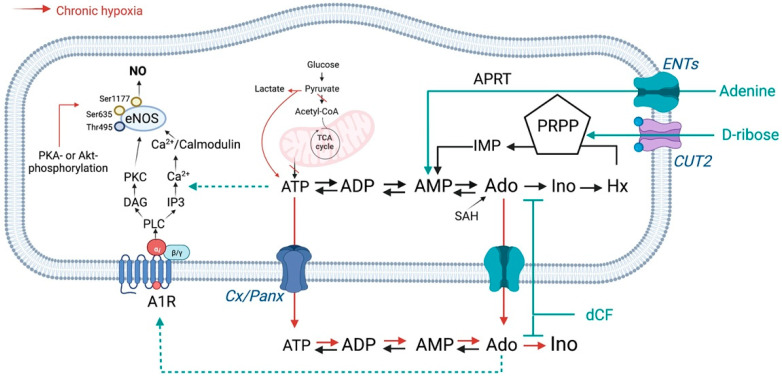
Pathways of nucleotide and adenosine metabolism in endothelial cells under hypoxia. The addition of adenine with ribose promotes both synthesis of adenine nucleotides and adenosine production, while 2′deoxycoformycin (dCF), decreasing adenosine to inosine catabolism inside and outside the cells, enhances the effect on adenosine receptor signaling. ATP, adenosine triphosphate; ADP, adenosine diphosphate; AMP, adenosine monophosphate; Ado, adenosine; Ino, inosine; Hx, hypoxanthine; APRT, adenine phosphoribosyltransferase; IMP, inosine monophosphate; PRPP, phosphoribosylpyrophosphate; SAH, S-adenosylhomocysteine hydrolase; Rib-1P, ribose-1-phosphate; A1R, adenosine A1 receptor; ENTs, equilibrative nucleoside transporters, CUT2, carbohydrate uptake transporter-2; Cx, connexins; Panx, pannexins; TCA cycle, tricarboxylic acid cycle; PLC, phospholipase C; IP3, inositol 1,4,5-trisphosphate; DAG, diacylglycerol; PKC, phosphokinase C; NO, nitric oxide; PKA, protein kinase A; Akt, serine/threonine protein kinase Akt. Created with https://BioRender.com (accessed on 27 April 2022).

## Data Availability

The data presented in this study are available on request from the corresponding author.

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
