# Peer review of "CoCl2-Mimicked Endothelial Cell Hypoxia Induces Nucleotide Depletion and Functional Impairment That Is Reversed by Nucleotide Precursors"

_biomedicines, 2022, doi:10.3390/biomedicines10071540_

Round 1

Reviewer 1 Report

The manuscript is well written. However, it may be enhanced by addressing the following points in greater depth:

- the signalling pathways induced by CoCl2, which has been used to induce hypoxia, have to be described;

- what are adenosine concentrations in hypoxic cells? The increase in adenosine concentrations in hypoxia is widely documented. What effect might adenosine concentrations have on NO modulation?

Author Response

Response to the Reviewer has been attached. Please see below. 

Reviewer 2 Report

The authors show HIF1a increased by CoCl2 treatment in human and mouse microvascular ECs. However, ATP and NO production were decreased in these ECs. Adenine and ribose supplementation enhanced intracellular ATP and NO production, which were further enhanced by adenosine signalling.

The study sounds interesting however there are still some questions needed to be addressed.

Major questions:

1, As we know, CoCl2 and hypoxia are not the same things.  CoCl2 can mimic hypoxia condition, which mainly reduces HIF degradation via affecting PHD and HRE. One of HIF protective effects is to slow down the reduction of ATP and protect cells from necrosis and apoptosis. The authors need to explain why CoCl2 depleted ATP in ECs.

2. NOSs are HIF target gens. CoCl2 increased HIF availability and subsequently NOSs expression from previous publications.  Why did CoCl2 decrease NO production?

3.  in Fig 1, CoCl2 increase HIF availability. Can this finding be shown in hypoxia condition not only the mimic? CoCl2 can inhibit HIF degradation via phd: is there any changes in HIF1b, HIF2 and HIF3?

4. in Fig 3 Adenine and ribose supplementation enhanced intracellular nucleotide pool and endothelial function: Do authors mean NO production is EC function? Is function of EC only to produce NO?

5 in Fig 4, total protein concentration is decrease by CoCl2 treatment. Is there any reason? By increased necrosis or apoptosis? The observation differs from Fig1.

6 In fig1 and 4, NO production was decreased by CoCl2 and the authors showed decreased eNOS in ECs in Fig6. How about the other isoforms of NOS?

7 In fig 2 and Fig5, activity of extracellular nucleotide hydrolysis and adenosine deamination was upregulated. Can this explain authors finding: intracellular ATP decrease.

8, Fig 7 shows the contribution of adenosine receptors including A1R, A2aR and A2bR in the protective effect by using their antagonist.  Why did not test antagonist A3R? As using antagonist can block the protective effect, can adding adenosine receptor agonists mimic the protective effect?

9 The authors show CoCl2 affects ATP/ADP production. Did P2 receptors contribute the alteration of NO production and which one(s)? e.g. P2X7, P2Y1, P2Y6 and P2Y13, which are involved in NO signalling?

10, In figs, authors should use CoCl2 instead of hypoxia. As we known, CoCl2 is not hypoxia. Change the hypoxia to CoCl2, please.

Author Response

(The authors gave the same response as above.)

Round 2

Reviewer 1 Report

The manuscript has been modified according to previous indications.

Reviewer 2 Report

Minor comments:

The authors replied the most questions however there are still issues needed to be addressed in the future:

1, authors suggested a study showing AR3R was not on EC but in that study aortic EC was studied not microvascular EC. They are not same things. Therefore AR3R should be tested.

2, authors suggested in the resting EC there is no iNOS. However in the current study, the EC in this study was stimulated by CoCl2.  iNOS is one of the most important target genes. Is there any alteration of iNOS in activity and expression under COCl2 treatment?

3, activation of purinergic receptor(s) affect NO signaling, especially phosphorylation of eNOS. In this study, ATP/ADP is one of major readouts so it is difficult to avoid to study the contribution of purinergic receptor(s).

This manuscript is a resubmission of an earlier submission. The following is a list of the peer review reports and author responses from that submission.